# A New Species *Nyctegretis seminigra* sp. nov. (Pyralidae, Phycitinae) Revealed by Congruent Morphological and Mitogenomic Evidence

**DOI:** 10.3390/insects16040413

**Published:** 2025-04-14

**Authors:** Linlin Yang, Yuxian Zhou, Yingdang Ren

**Affiliations:** 1Henan Key Laboratory of Agricultural Pest Monitoring and Control, Key Laboratory of Integrated Crop Pests Management on Crops in Southern Region of North China, Ministry of Agriculture and Rural Affairs, Institute of Plant Protection, Henan Academy of Agricultural Sciences, Zhengzhou 450002, China; yangll_tineidae@163.com (L.Y.); zhouyx_z@163.com (Y.Z.); 2School of Agriculture, Ningxia University, Yinchuan 750021, China

**Keywords:** mitogenome, morphology, new species, *Nyctegretis*, phylogeny, snout moths

## Abstract

As the largest subfamily within Pyralidae, Phycitinae comprises over 3300 known species worldwide. The considerable diversity and high incidence of homoplasy within Phycitinae present significant challenges for species identification when relying exclusively on morphological characters. Therefore, the integration of molecular data analysis is particularly crucial. In this study, we describe a new pyralid moth species, integrating both morphological and molecular evidence. We provide detailed illustrations of adults, wing venation, male and female genitalia, DNA barcodes, and the complete mitochondrial genome of the new species, and conduct a phylogenetic analysis based on mitochondrial genes to get a better understanding of the taxonomic status of the new species.

## 1. Introduction

The genus *Nyctegretis* was established by Zeller [1], with *Tinea achatinella* Hübner designated as its type species. Roesler [2] systematically revised the Palaearctic species of *Nyctegretis* and synonymized *Synallorema* Gozmány and *Trichorachia* Hampson under the genus. Subsequently, Leraut [3,4] further synonymized *Pseudopiesmopoda* Roesler and *Mesciniadia* Hampson with *Nyctegretis*. It is characterized by forewing with stalked veins R_3_+R_4+5_, hindwing with fused M_2_ and M_3_, male genitalia featuring a narrow band-like transtilla, and female genitalia with a well-developed antrum and a signum formed by chitinous tubercles. Currently, the genus comprises 11 species [5], with a remarkably broad yet disjunct global distribution spanning the Afrotropical, Australasian, and Palaearctic regions. In China, two species have been recorded [6].

Mitogenomes have been used for phylogenetic analysis in several Lepidoptera groups [7,8]. In Phycitinae, mitogenomes of 25 species from 16 genera have been sequenced (GenBank, https://www.ncbi.nlm.nih.gov, accessed on 9 January 2025). However, most of these sequences lack detailed descriptions. As the largest subfamily within Pyralidae, Phycitinae comprises over 3300 known species worldwide [4,9]. It is obvious that the current mitochondrial genome data are insufficient to adequately represent such a diverse group.

In this study, we describe a new species, *N. seminigra* sp. nov., based on specimens collected from Guangxi, Hainan, and Yunnan, China. Diagnosis and illustrations of the adults and genitalia are provided for the new species. The complete mitogenomes of *N. seminigra* sp. nov. and *N. triangulella* are sequenced and annotated here. This study represents the first report on the mitochondrial genome data for this genus. These findings will contribute to a better understanding of the genus *Nyctegretis*, and provide valuable data for further phylogeny-based systematic studies.

## 2. Materials and Methods

### 2.1. Sample Collection

The specimens examined in this study were collected using 250 W high-pressure mercury lamps on a white sheet. Samples for DNA extraction were preserved in absolute ethanol and stored at −20 °C. The type specimens of *N. seminigra* sp. nov. are deposited in HAASM and NKU. Detailed depository information for each specimen is indicated in the systematic section.

### 2.2. Morphological Analyses

Genitalia were dissected according to Li [10]. Images of adults and genitalia were taken using Leica M205A and Leica DM750 microscopes, respectively, coupled with Leica Application Suite 4.6 software (Leica, Wetzlar, Germany). Morphological terminology follows Roesler [2] and Slamka [11].

### 2.3. DNA Extraction and Sequencing

DNA extraction and mitochondrial *cox1* gene amplification were performed following our previous study [12]. The final 658 bp barcode region of *cox1* was registered in GenBank (accession numbers: PV156261–PV156270). Two alcohol-preserved specimens were sent to BGI Tech Solutions Co., Ltd. (Shenzhen, China) for DNA extraction and library preparation. Genome sequencing was conducted on the DNBSEQ platform using 150 bp paired-end reads to generate high-quality sequencing data. The complete mitogenomes of *N. seminigra* sp. nov. and *N. triangulella* have been submitted to GenBank (accession numbers: PV021108 and PV021107, respectively).

### 2.4. DNA Barcode Analysis

Seventy-one *cox1* sequences from six species of *Nyctegretis* were used to calculate interspecific and intraspecific divergences. Ten of these sequences were generated using Sanger sequencing, while two sequences were derived from mitochondrial genome data obtained through next-generation sequencing in this study, and the remaining sequences were retrieved from the BOLD System. The BIN numbers are as follows: *N. cullinanensis* (AEX2771, AEW9294); *N. infractalis* (AAX3960); *N. lineana* (AAE5583); *N. triangulella* (ACN1315); *N. ruminella* (ACJ7106). Genetic distances were calculated under the Kimura 2-parameter model [13] using MEGA X.

### 2.5. Mitogenome Analysis

Complete mitogenomes were assembled using MitoZ v3.6 [14]. Annotations were performed using Geneious Prime R11 [15] and MITOS (Galaxy 24.2.rc1) [16]. Nucleotide composition, amino acid usage frequency, and relative synonymous codon usage (RSCU) of the protein-coding genes (PCGs) were calculated using PhyloSuite v1.2.3 [17,18]. Mitogenome maps were depicted using Blast Ring Image Generator (BRIG) [19]. Fifty-nine mitogenomes of Pyralidae (as ingroups) and two mitogenomes of Crambidae (as outgroup sequences) were used for phylogenetic analysis using PhyloSuite v1.2.3.

## 3. Results

### 3.1. Systematics

#### 3.1.1. Genus *Nyctegretis* Zeller, 1848

Type species: *Tinea achatinella* Hübner, [1824], by monotypy.

*Mesciniadia* Hampson in Ragonot and Hampson, 1901. Type species: *Nephopteryx* [sic] *infractalis* Walker, 1864, by monotypy.

*Trichorachia* Hampson, 1930. Type species: *Trichorachia leonine* Hampson, 1930, by monotypy.

*Synallorema* Gozmány, 1958. Type species: *Nyctegretis triangulella* Ragonot, 1901, by monotypy.

*Pseudopiesmopoda* Roesler, 1982. Type species: *Pseudopiesmopoda malgassicola* Roesler, 1982, by monotypy.

Diagnosis. Species of the genus *Nyctegretis* share several characteristic head features with those of other genera in the tribe Phycitini: the head is covered with rough scales; the male antenna bears appressed scales dorsally and is pubescent ventrally, lacking sinus in base of the flagellum, while the female antenna is filiform; the labial palpus is slender throughout, extending upward beyond the vertex; and the maxillary palpus is short and slightly flattened. However, *Nyctegretis* can be distinguished from related genera by its wing venation and structures of the male and female genitalia: the forewing has long-stalked R_3_ and R_4+5_ and stalked or fused M_2_ and M_3_; the hindwing has fused M_2_ and M_3_ shortly stalked with CuA_1_; the transtilla is narrow and band-like and the relatively wide valva has a rod-like costa that is usually protruding at the end in the male genitalia; the antrum is usually well developed, and the signum (absent in *N. aenigmella*) consists of a group of triangular or plate-like chitinous tubercles in the female genitalia.

Biology. The larvae of *Nyctegretis lineana* feed on a variety of plants, including *Ononis*, *Artemisia*, *Antennaria*, *Gnaphalium*, *Sedum*, *Cytisus*, and *Trifolium* [20]. The larvae of *N. ruminella* feed on various detritus rather than on green leaves [21]. The host plants of the remaining species within this genus are currently unknown.

Remarks. Gozmány [22] established the genus *Synallorema* based on the monotypic species *Nyctegretis triangulella* Ragonot (originally described by Ragonot in Ragonot and Hampson, 1901: 29 [23]), which was known to be found in Japan, Hungary, and eastern Austria. Roesler [2,24] treated *Synallorema* as a synonym of *Nyctegretis*, a view followed by Du et al. [6]. However, Fletcher and Nye [25] mistakenly treated *Homoeosoma triangulella* Hampson (described by Hampson in Ragonot and Hampson, 1901: 256) from “Sou-tcheou” (Suzhou, Jiangsu Province, China) as its type species and thereby regarded *Synallorema* to be a valid genus name. In agreement with Roesler’s perspective, we hereby treat *Synallorema* as a synonym of *Nyctegretis*.

Checklist and Distributions of the Species in *Nyctegretis*

1.*Nyctegretis aenicta* (Turner, 1913) [26]
*Ecbletodes aenicta* Turner, 1913: 120. Type locality: Australia (Northern Queensland).Distribution: Australia.
2.*Nyctegretis aenigmella* Leraut, 2002 [27]
*Nyctegretis aenigmella* Leraut, 2002: 148. Type locality: France (Corse).Distribution: France.
3.*Nyctegretis cullinanensis* Balinsky, 1991 [28]
*Nyctegretis cullinanensis* Balinsky, 1991: 111. Type locality: South Africa (Transvaal).Distribution: Kenya, South Africa.
4.*Nyctegretis inclinella* Ragonot, 1888 [29,30,31]
*Nyctegretis inclinella* Ragonot, 1888: 32. Type locality: South Africa (Eastern Cape).Distribution: Madagascar, Mozambique, South Africa.
5.*Nyctegretis infractalis* (Walker, 1864) [32,33]
*Nephopteryx infractalis* Walker, 1864: 958. Type locality: Malaysia (Borneo).Distribution: Australia, Indonesia, Malaysia.
6.*Nyctegretis leonina* (Hampson, 1930) [34]
*Trichorachia leonina* Hampson, 1930: 65. Type locality: Sierra Leone, Zimbabwe (South Rhodesia).Distribution: Sierra Leone, South Africa, Zimbabwe.
7.*Nyctegretis lineana* (Scopoli, 1786) [2,6,35,36,37,38]
*Phalaena lineana* Scopol, 1786: 57. Type locality: Italy (Insubria).*Tinea achatinella* Hübner, 1824: pl. 68. Type locality: Europe.*Nyctegretis achatinella* var. *griseella* Caradja, 1910: 130. Type locality: France (Dax, Vernet), Kazakhstan (Oral).*Nyctegretis calamitatella* Roesler, 1973: 290. Type locality: China (Shanxi).*Nyctegretis lineana katastrophella* Roesler, 1970: 47. Type locality: Mongolia (Cojbalsan aimak)Distribution: Austria, China, England, Finland, Germany, Italy, Japan, Korea, Mongolia, Netherlands, Norway, Russia, and Spain.
8.*Nyctegretis malgassicola* (Roesler, 1982) [3,31,39]
*Pseudopiesmopoda malgassicola* Roesler, 1982: 858. Type locality: Madagascar.*Nyctegretis malgassicola insularis* Leraut, 2019: 38. Type locality: Comoros (Grande Comore).Distribution: Comoros, Madagascar, Mascareignes.
9.Nyctegretis otoptila (Turner, 1913) [26]
*Ecbletodes otoptila* Turner, 1913: 120. Type locality: Australia (Northern Territory).Distribution: Australia.
10.*Nyctegretis ruminella* La Harpe, 1860 [2,21,35,40]
*Nyctegretis ruminella* La Harpe, 1860: 397. Type locality: Italy (Sicily).Distribution: Bulgaria, France, Gibraltar, Greece, Italy, Malta, North Africa, Romania, Russia, Spain, and Turkey.
11.*Nyctegretis triangulella* Ragonot, 1901 [2,6,35,36,37,38]
*Nyctegretis triangulella* Ragonot, in Ragonot and Hampson, 1901: 29. Type locality: Japan.*Nyctegretis impossibilella* Roesler, 1969: 205. Type locality: Greece (Platamon).Distribution: Austria, China, Czech Republic, England, Germany, Greece, Hungary, Iraq, Iran, Israel, Italy, Japan, Romania, Russia, Slovakia, and Turkey.
12.*Nyctegretis seminigra* Yang, Zhou and Ren, sp. nov.
Type locality: China (Guangxi, Hainan, Yunnan).Distribution: China (Guangxi, Hainan, Yunnan).


Key to the species of *Nyctegretis*

1. Forewing with distinct antemedial line and postmedial line …………………………….2-  Forewing with antemedial line and postmedial line indistinct or absent ……………....72. Underside of male forewing with oblique ridge of long hair from origin of R_4_ to M_1_  before termen [34] …………………………………………………………………..*N. leonina*-  Underside of male forewing without oblique ridge of long hair ……………….………..33. Signum band-like ……………………………………………………………………………...4-  Signum raindrop-shaped or oval …………………………………………………………….64. Costa of valva without protrusion [30,31] ……………………………………...*N. inclinella*-  Costa of valva with a protrusion ………………….…………………………………………55. Subapical protrusion on costa of valva small and pointed; signum located in the  anterior half of the corpus bursa ……………………………………..*N. seminigra* sp. nov.-  Subapical protrusion on costa of valve large, split into a stocky extension and a curved    spine; signum located in the posterior half of the corpus bursa [2,35] …...*N. triangulella*6. Uncus rounded at apex; signum raindrop-shaped [2,21] ……….…………...*N. ruminella*-  Uncus narrowed and pointed at apex; signum oval [2,41] ……………………...*N. lineana*7. Forewing with a white costal band …………………………………………………………8-  Forewing without white costal band …………………………………….………………...108. Forewing with basal and external area yellow, medial area dark brown; signum   absent [27] ………………………………………………………………………..*N. aenigmella*-  Forewing gray or yellowish brown; signum present ……………………………………...99. Costa of valva clubbed; signum triangular [28] …………………………..*N. cullinanensis*-  Costa of valva bifurcated; signum oval [39];………………………………..*N. malgassicola*10. Hindwing normal, not arched at base of costa [26] …….……………………..*N. aenicta*-  Hindwing arched at base of costa, with a tuft of long hair ……………………………..1111. Forewing with CuA_1_ stalked with CuA_2_ [26] ………………………………....*N. otoptila*-  Forewing with CuA_1_ separated from CuA_2_ [33] ……………………………..*N. infractalis*

#### 3.1.2. *Nyctegretis seminigra* Yang, Zhou, and Ren, sp. nov.

Zoobank: urn:lsid:zoobank.org:act:CA7E2E67-D432-47B0-88B3-679C987760BD

Type material. Holotype ♂, CHINA, Hainan Province, Wuzhishan, alt. 580 m, 11.IV.2013, leg. Ying-Dang Ren, Xiao-Guang Liu (HAASM). Paratypes: Hainan Province: 8♂1♀, same data as holotype, except dated 11–12. IV.2013 (HAASM); 1♀, Wuzhishan, alt. 700 m, 19.V.2007, leg. Zhi-Wei Zhang, Wei-Chun Li, genitalia slide No. LJY11392; 3♂, Jianfengling, alt. 960 m, 13–15.IV.2013, leg. Ying-Dang Ren, Xiao-Guang Liu (HAASM); 1♂, Diaoluoshan, alt. 900 m, 9.IV.2013, leg. Ying-Dang Ren, Xiao-Guang Liu (HAASM); 1♂, Wuzhishan City, Shuiman Town (18.90 N, 109.67 E), alt. 735 m, 25.VIII.2020, leg. Linlin Yang, genitalia slide No. DNAYLL18393 (HAASM); 1♂, Yinggeling, alt. 620 m, 28.III.2010, leg. Bingbing Hu, genitalia slide No. LJY10453. Guangxi Zhuang Autonomous Region: 1♀, Chongzuo City, Longzhou County, Pona, Nonggang Nature Reserve (22.49 N, 106.95 E), alt. 160 m, 19.VIII.2020, leg. Linlin Yang, genitalia slide No. DNAYLL18388 (HAASM). Yunnan Province: 3♂3♀, Honghe Prefecture, Pingbian County, Daweishan, alt. 1800 m, 6.XI.2010, leg. Bingbing Hu et al., genitalia slide Nos. DNAYLL18046m, WYQ14041f (NKU); 1♂, Puer City, Yunpanshan Town, Taiyanghe Nature Reserve, alt. 1450 m, 2.ix.2014, leg. Zhenguo Zhang, genitalia slide No. DNAYLL18041 (NKU); 1♀, Puer City, Yunpanshan Town, Taiyanghe Nature Reserve, alt. 1450 m, 3.VI.2014, leg. Zhenguo Zhang (NKU); 1♂, Puer City, Yunpanshan Town, Taiyanghe Nature Reserve, alt. 1580 m, 6.viii.2020, leg. Linlin Yang, genitalia slide No. DNAYLL18399 (HAASM).

Diagnosis. Externally, the new species can be easily distinguished from its relatives by the bicolored forewing pattern. The genitalia structures of the new species closely resemble those of *N. triangulella*, but can be differentiated from the latter by the costa of the valva with a small, pointed protrusion subapically in the male, and the signum located in the anterior half of the bursa in the female. In *N. triangulella*, the costa of the valva is unequally split at the terminal end into a stocky extension and a narrowly curved spine in the male genitalia, and the signum is located in the posterior half of the bursa in the female genitalia.

Description. Adult (Figure 1). Wingspan 14.0 mm in holotype, 13.5–15.0 mm in male paratypes, 14.5–17.5 mm in female paratypes.

*Head* (Figure 1b,c). Frons with appressed pale gray scales, shining silvery; vertex grayish brown, with dark-tipped, rough, forward-directed scales; ochreous-yellowish occiput. Antenna: scape 1.5-times as long as wide, with a blackish brown dorsal surface, pale yellow ventral surface; yellowish-brown flagellum, scales tipped with black. Labial palpus curved upwards, reaching above the vertex, segmental ratio about 1:1.8:1 in males, 1:1.5:1.2 in females; first segment pale yellow, tinted with a few dark scales, terminal two segments dark brown but grayish yellow at distal ends. Maxillary palpus yellowish brown, about 1/4 length of labial palpus, three-segmented, of nearly equal length. *Thorax*. Patagium, tegula, and thorax bronze, individual scales dark-tipped. Foreleg dark brown, except yellowish brown at apices of tarsomeres; mid- and hindleg predominantly grayish yellow, interspersed with blackish-brown markings, tarsomeres dark brown, margined with grayish white at apices.

*Wings* (Figure 1a,d). Forewing with a bicolored pattern: basal area grayish yellow, with a few scattered blackish brown scales at base, along costa, and dorsum; medial and external areas blackish brown, faintly tinged with grayish yellow; antemedial line grayish yellow, hardly distinguishable from basal area, extending from near basal 2/5 of costa outwardly oblique to distal 2/5 of dorsum; postmedial line grayish yellow, slightly dentate and sinuate, extending from distal 1/4 of costa to distal 1/5 of dorsum; discal spots confluent into a grayish yellow line; terminal line yellowish brown; cilia dark gray. Hindwing scales grayish brown to dark gray; cilia grayish brown. Venation: forewing with R_3_ and R_4+5_ stalked for about 2/3 of their total length, R_2_ stalked R_3_+R_4+5_ about half length of R_2_, M_2_, and M_3_ stalked at basal 1/4 and terminating at the same point; hindwing with Sc+R_1_ and Rs stalked for about half length of Sc+R_1_, M_2_ and M_3_ fused and shortly stalked with CuA_1_.

*Male genitalia* (Figure 1e). Uncus triangular, length approximately 1.3 times its width, tapered to a slightly pointed apex, densely hirsute on distal third. Gnathos 3/4 length of uncus, of nearly equal width throughout, rod-like, truncate at apex. Transtilla slender, arched, band-like. Valva about three times length of width, narrow at base, gradually broadened to rounded apical margin; costa not reaching to the end of the valva, clubbed, with a pointed protrusion subapically; sacculus about 2/5 length of valva; a clasper-like, elevated chitin structure beyond apex of sacculus; Vinculum rounded trapezoidal, length about 2/3 of greatest width. Juxta U-shaped, middle part rounded quadrilateral, lateral lobes slender, fingerlike, about 2/3 length of gnathos. Aedeagus clavate, about 4/5 length of valva, base slightly broader than apex; with a thorn-like cornutus. Culcita one pair, simple, about 4/5 length of valva.

*Female genitalia* (Figure 1f). Papillae anales subtriangular, approximately twice as long as its width, apex rounded, sparsely covered with long setae. Eighth tergite about 3/4 as long as its width, anterior margin slightly concave in middle, posterior margin straight. Apophysis anterioris 1.3-times length of apophysis posterioris. Ostium bursae rather wide. Antrum strongly sclerotized, cup-shaped, approximately 3/4 as long as its width. Ductus bursae membraneous, gradually expanding into corpus bursae. Corpus bursae about 1.6-times as long as its width, elliptically rounded, densely covered with granules at the junction with ductus bursae. Signum represented by a band-like structure composed of numerous plate-like chitinous tubercles arranged obliquely, located in the anterior 1/5 to 1/2 region of corpus bursae. Ductus seminalis arising from near middle of corpus bursae, approximate to signum.

Biology. Larval host plant is unknown.

Distribution. China (Guangxi, Hainan, Yunnan).

Etymology. The species name is derived from the Latin prefix *semi-* (meaning half), and the Latin word *nigrum* (meaning black), referring to the forewing of the new species that has a blackish brown distal half (including the medial and external areas).

### 3.2. DNA Barcode Analysis

The genetic distance among six species of *Nyctegretis* was determined through pairwise analysis of 71 sequences. The results (Table 1) indicate that all included species possess highly species-specific DNA barcodes. The minimum interspecific divergences to the nearest species ranged from 7.16% to 9.82%, while the maximal intraspecific distances ranged from 0.46% to 2.58%. For *N. seminigra* sp. nov., the intraspecific genetic divergence among the six barcoded specimens ranged from 0 to 1.26%. Its genetically closest relatives are *N. lineana* and *N. triangulella*, with interspecific divergences of 9.66% to 10.36% and 9.73% to 11.71%, respectively.

### 3.3. Mitogenome Analysis

The mitochondrial genome is 15,198 bp-long in *N. seminigra* sp. nov. and 15,205 bp-long in *N. triangulella*. The structure and composition of the mitochondrial genomes in both species are consistent with those of other pyraloid moths, containing a total of 37 genes: 13 PCGs, 22 tRNA genes, two rRNA genes; and one A+T rich region (Figure 2; Appendix A). Among these genes, 23 are located on the majority strand (J-strand), whereas the remaining 14 genes are encoded on the minority strand (N-strand). Both mitogenomes are compact, with no gene rearrangements or gene losses. In *N. seminigra* sp. nov., there are eight gene overlaps ranging from 1 bp to 8 bp in length, with the largest overlap located between *trnW* and *trnC*. In contrast, *N. triangulella* exhibits ten gene overlaps ranging from 1 bp to 12 bp in length, and the largest overlap is found between *rrnL* and *trnV*. *N. seminigra* sp. nov. contains 14 gene spacers ranging from 2 bp to 44 bp in length, while *N. triangulella* has 12 intergenic spacers ranging from 2 bp to 42 bp in length. The largest intergenic spacer in both species is located between *trnQ* and *nad2*, consistent with observations from other studies on pyraloid moths [42,43].

#### 3.3.1. Nucleotide Composition

The base composition of the genes in the mitogenomes of *N. seminigra* sp. nov. and *N. triangulella* is summarized in Appendix A. The A+T content in the entire nucleotide sequence is 79.2% for *N. seminigra* sp. nov. and 80.4% for *N. triangulella*, indicating a clear AT bias in both species. This bias is consistent across all genetic elements. Both species exhibit negative AT skewness and negative GC skewness values throughout their entire sequences, suggesting a higher abundance of T over A and C over G. This pattern is consistent with the base composition observed in other groups within the superfamily Pyraloidea. In both species, the PCGs and A+T rich regions display negative AT skewness values, indicating a preference for T over A. In contrast, the tRNA and rRNA gene regions show positive AT skewness values, revealing a preference for A over T. The GC skewness values for the PCGs, tRNA genes, and rRNA genes are positive in both species, indicating a preference for C over G in these regions.

#### 3.3.2. PCG Regions and Codon Usage

The combined lengths of PCGs in the two mitogenomes are 11,184 bp (*N. seminigra* sp. nov.) and 11,190 bp (*N. triangulella*), respectively (Appendix A). Comparative analysis revealed that the majority of individual genes exhibited near-identical lengths between the two species, with one exception observed in *nad2*, displaying lengths of 1008 bp in *N. seminigra* sp. nov. mitogenome compared to 1014 bp in *N. triangulella* mitogenome. In both mitogenomes, *atp8* is the shortest gene while *nad5* is the longest (Appendix A).

Comparative analysis of start and stop codon usage revealed consistent and divergent patterns between the two mitochondrial genomes. Most of the start codons are consistent between the two mitogenomes, with ATG used in five PCGs (*atp6*, *cox3*, *nad4*, *nad4L*, *cytb*), and ATT in another five PCGs (*nad2*, *cox2*, *atp8*, *nad3*, *nad5*); CGA is used as the start codon for *cox1* and ATA is used for *nad1*. The start codon for *nad6* differs between the *N. seminigra* sp. nov. mitogenome (ATA) and *N. triangulella* mitogenome (ATT). The most common identical stop codon in both the mitogenomes is TAA (in six PCGs: *nad2*, *atp8*, *atp6*, *cox3*, *nad4L*, *cytb*), followed by two incomplete T stop codon (*cox2*, *nad5*), one incomplete TA stop codon for *nad4*, and one TAG for *nad3*. For the remaining three PCGs, the stop codons differ between the two mitogenomes: *nad6* and *nad1* use TAG as their stop codon, and *cox1* uses TAA in the *N. seminigra* sp. nov. mitogenome; meanwhile, *nad6* and *nad1* use TAA and *cox1* has an incomplete T stop codon in the *N. triangulella* mitogenome.

The amino acids’ frequency and relative synonymous codon usage (RSCU) values for PCGs in the two mitogenomes are summarized in Appendix A and Appendix A. The most common amino acid in both mitogenomes is Ile, with a frequency of 12.21% in the *N. seminigra* sp. nov. mitogenome, and 12.31% in the *N. triangulella* mitogenome. The least frequently used amino acid is Cys, with respective frequencies of 0.78% and 0.86%. Codon-level analysis demonstrated a significant AT bias, evidenced by a preferential usage of NNA and NNU codons across both mitogenomes. In both the mitogenomes, 26 codons exhibit positive codon usage bias (RSCU > 1), indicating that these codons are used more frequently. In contrast, 36 codons in the *N. seminigra* sp. nov. mitogenome and 35 codons in the *N. triangulella* mitogenome show a negative codon usage bias (RSCU < 1). Among these, two codons, CUG (Leu1) and CGC (Arg), are used only in the *N. seminigra* sp. nov. mitogenome, while CCG (Pro) is used only in the *N. triangulella* mitogenome; one codon, AGG (Ser1) is never used in either mitogenome.

#### 3.3.3. tRNA Genes, rRNA Genes, and the A+T Rich Region

The mitochondrial tRNA genes are AT-rich, with *N. seminigra* sp. nov. mitogenome exhibiting a 80.4% AT-content versus 81.3% in the *N. triangulella* mitogenome, accompanied by positive AT-skew (0.011 and 0.015) and GC-skew (0.184 and 0.166) values, respectively (Appendix A). The relative positions and anticodon sequences of tRNAs in both mitogenomes are identical, spanning 63–71 bp in length. The cloverleaf secondary structures (Appendix A) of the tRNA genes are generally similar between the two mitogenomes. Both mitogenomes lack the DHU arm in *trnS1* and TΨC loop in *trnI*. In addition to the typical Watson–Crick base pairs (A-U and G-C), a small number of U-U mismatches are observed in the TΨC arm of *trnE* and anticodon arm of *trnS2* in both mitogenomes. The *N. seminigra* sp. nov. mitogenome exhibits additional U-U mismatches localized in the anticodon arm of *trnQ* and *trnR*.

The mitochondrial rRNA genes are also AT-rich: 84.5% in the *N. seminigra* sp. nov. mitogenome, and 84.6% in the *N. triangulella* mitogenome; both genes show positive AT-skew (0.071 and 0.091) and GC-skew (0.368 and 0.363) values (Appendix A). The two rRNA genes (*rrnL* and *rrnS*) are encoded on the N-strand and are separated by *trnV* (Figure 2). The *rrnS* gene (774 bp) and *rrnL* gene (1372 bp) in the *N. seminigra* sp. nov. mitogenome are slightly shorter than those in the *N. triangulella* mitogenome (781 bp and 1390 bp, respectively).

The A+T rich region is the longest noncoding region in both mitogenomes, with a length of 320 bp in *N. seminigra* sp. nov. mitogenome and 312 bp in *N. triangulella* mitogenome. Characterized by its high AT content, this region also displays a negative AT skew and GC skew.

#### 3.3.4. Phylogenetic Analysis Based on Mitogenomes

A phylogenomic framework (Figure 3) was reconstructed using concatenated mitogenomes from two newly sequenced specimens and 57 publicly available sequences, analyzed through Maximum Likelihood (ML) and Bayesian Inference (BI) approaches. Phycitinae is confirmed as a robustly supported monophyletic lineage (BS = 100, PP = 1.00), which bifurcates into two major clades, with Clade A (BS = 93, PP = 1.00) exhibiting stronger nodal support than Clade B (BS = 74, PP = 1.00). The monophyly of each genus is well supported, although intergeneric relationships remain poorly resolved. The phylogenetic position of the new species is highly supported in both analyses (BS = 100, PP = 1.00), which is consistent with morphological evidence and confirms its classification within the genus *Nyctegretis*.

## 4. Discussion

The new species is classified within the genus *Nyctegretis* based on the following morphological characteristics: the forewing with stalked R_3_ + R_4+5_ and stalked M_2_ + M_3_, the hindwing with M_2_ and M_3_ fused and stalked with CuA_1_; the transtilla is characterized as a narrow band, and the broad valva has a rod-like costa that protrudes at its end in the male genitalia; the antrum is well-developed and the signum comprises a cluster of plate-like chitinous tubercles in the female genitalia. The results of the molecular phylogenetic analysis further corroborate this classification, demonstrating that mitochondrial genome data are valuable resources for species identification and taxonomic status determination.

It should be noted that some characteristics of *Nyctegretis* are also observed in other phycitines, and there are individual species within *Nyctegretis* that do not adhere to the genus characteristics (e.g., *N. aenigmella* lacks a signum; *N. infractalis* with stalked CuA_1_ + CuA_2_ in the forewing). This is common in Phycitinae. As indicated by Roe et al. [44], the considerable diversity within this subfamily, coupled with the high incidence of homoplasy (similar traits evolve independently in different lineages), poses great challenges for reconstructing phylogenetic relationships based solely on morphological features. In this case, the integration of molecular data analysis becomes particularly crucial. Roe et al. [44] provided the first molecular phylogeny of phycitine genera, resulting in the division of the subfamily into two clades. Similar results were obtained in later studies on the molecular phylogeny of Pyraloidea [9,45,46], including findings in this study. It is noteworthy, however, that these studies did not comprehensively sample most taxa of Phycitinae, focusing mainly on species from Phycitini. Only one species of Anerastiini (*Peoria approximella*) was included in the analysis and it is nested within Phycitini [44]. Additional representatives will need to be sampled to achieve a comprehensive molecular phylogenetic analysis.

## Figures and Tables

**Figure 1 insects-16-00413-f001:**
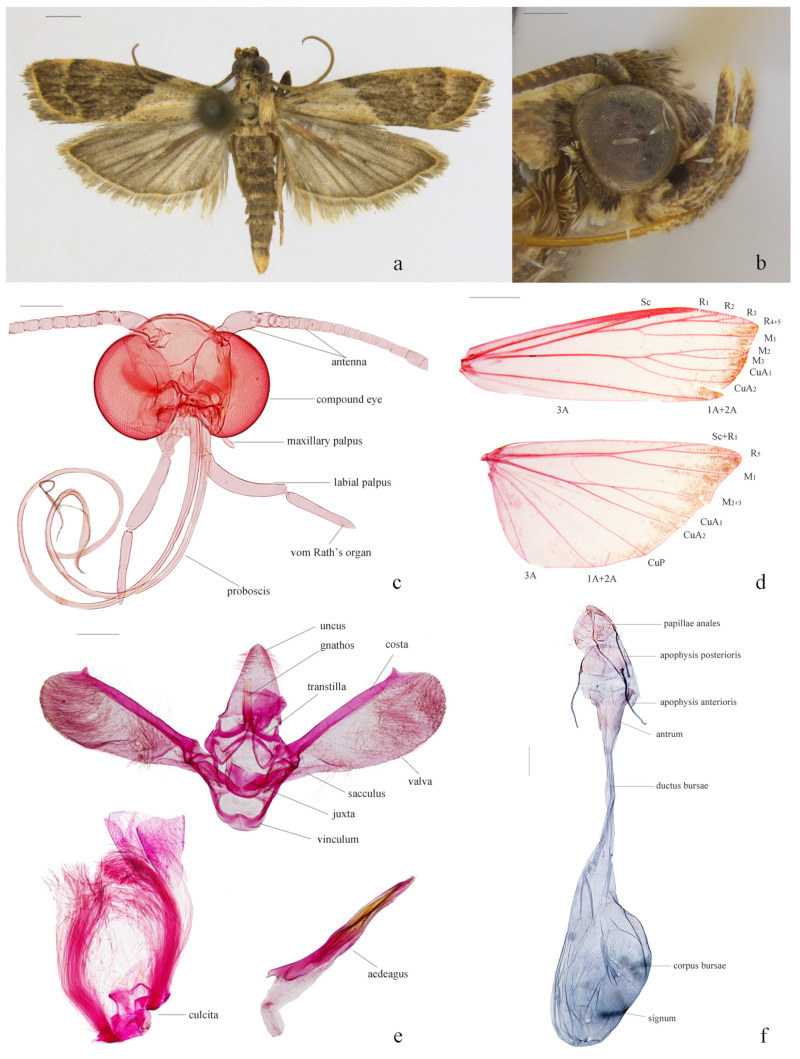
Morphological features of *Nyctegretis seminigra* sp. nov. (**a**), adult, holotype, male; (**b**), head in lateral view, paratype, male; (**c**), dissected head, paratype, female; (**d**), wing venation, paratype, female; (**e**), male genitalia, paratype; (**f**), female genitalia, paratype. ((**c**,**d**,**f**), slide No. DNAYLL18393; (**e**), slide No. LJY10453; (**a**), scale bar = 1 mm; (**b**–**e**), scale bars = 0.25 mm).

**Figure 2 insects-16-00413-f002:**
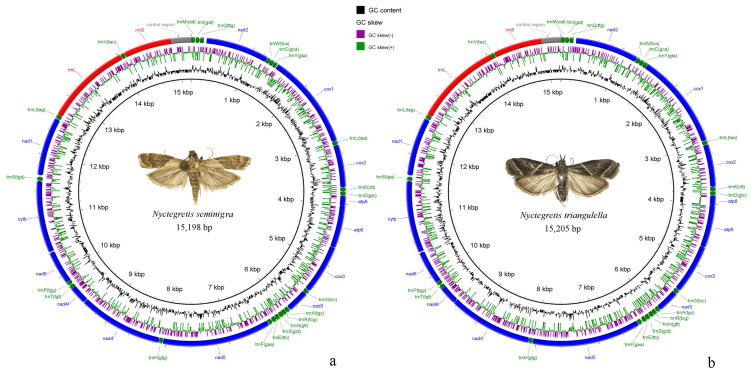
Complete mitogenome maps. (**a**), *Nyctegretis seminigra* sp. nov.; (**b**), *N. triangulella.* Arrows indicate the orientation of gene transcription. GC skew is shown on the outer surface of the ring whereas GC content is shown on the inner surface. The anticodon of each tRNA gene is shown in parentheses.

**Figure 3 insects-16-00413-f003:**
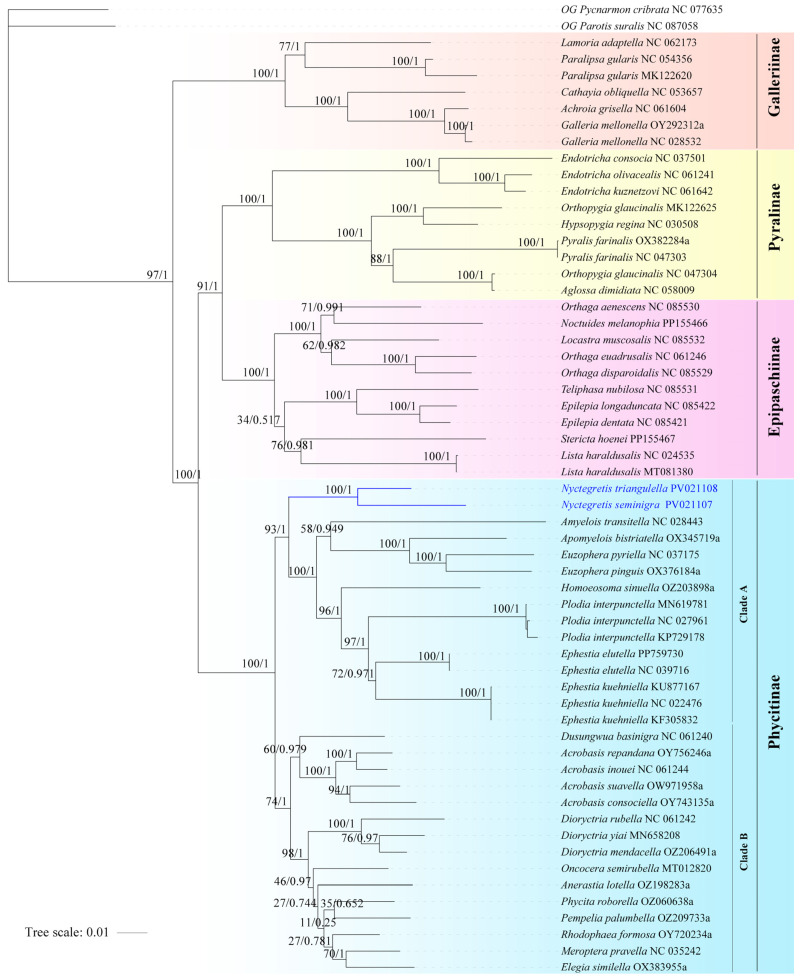
Maximum likelihood (ML) and Bayesian inference (BI) phylogenetic tree of Pyralidae based on mitogenomes (PCG dataset). Numbers near branches refer to bootstrap support values and Bayesian posterior probabilities. Species names are followed by their corresponding GenBank accession numbers.

**Table 1 insects-16-00413-t001:** Pairwise distance matrix of DNA barcodes calculated among seven species of *Nyctegretis*.

Species	1	2	3	4	5	6
1 *N. seminigra* sp. nov.	**0–1.26**					
2 *N. cullinanensis*	14.59–16.16	**0–2.58**				
3 *N. lineana*	9.66–10.36	11.18–12.75	**0–1.94**			
4 *N. ruminella*	11.33–11.89	11.89–13.22	8.73–9.12	**0–0.46**		
5 *N. triangulella*	9.73–11.71	10.72–12.63	7.16–8.65	7.41–9.50	**0–1.25**	
6 *N. infractalis*	10.82–12.52	9.82–11.55	7.64–8.56	9.45–10.33	8.76–10.30	**0.09–1.89**

Note: Genetic distances (%) were calculated with the Kimura two-parameter (K2P) substitution model using MEGA X; extreme values of intraspecific and interspecific distances are given (the numbers in bold are the intraspecific distances).

## Data Availability

DNA barcodes (PV156261–PV156270) and mitogenomes (PV021107–PV021108) generated in this study have been deposited in GenBank. Type specimens are deposited at HAASM and NKU. Publicly archived datasets used in this study were retrieved from the BOLD System (see materials and methods section for BIN numbers) and GenBank (see Figure 3 for accession numbers).

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
