# Peer review of "A New Species Nyctegretis seminigra sp. nov. (Pyralidae, Phycitinae) Revealed by Congruent Morphological and Mitogenomic Evidence"

_insects, 2025, doi:10.3390/insects16040413_

Round 1
Reviewer 1 Report
Comments and Suggestions for Authors
A detailed description of a new species, including mitogenomic data, is welcome in principle. However, phylogenetic analysys using a limited and largerly random set of species is unlikely to be appropriate in a paper devoted to the description of a single new species; it seems unnecessary and somewhat premature.
In the text, the type species of the genus Nyctegretis is designated as achatinella (page 1), and the distribution is given for lineana (page 2); nowhere is it noted that achatinella is a junior synonym of lineana.
On page 3 (section 3.1.1) the full generic name Nyctegretis Zeller, 1848 is given three times in a row; at least in the second case it is clearly redundant.
The specific epithet of the new species is formed incorrectly. Semi- + niger- should be transformed to seminigra or seminigrella (half-black), whereas seminigera in translation from Latin to English means half-breed (half-blood).
The article does not provide a decoding of the abbreviations of the depositories of type specimens.
The distribution of some Palearctic species of the genus Nyctegretis (lineana, triangulella) is given incompletely, without taking in account published data on different regions of Russia (see Sinev S.Yu., Streltzov A.N., Trofimova T.A. Pyralidae. - In: Sinev S.Yu. (ed.). Catalogue of the Lepidoptera of Russia. Edition 2. St. Petersburg: Zoological Institute RAS, 2019. P. 165-178.).
The text needs minor revision in accordance with the above comments.
Comments on the Quality of English LanguageThe reviewer is not a native English speaker and therefore refrains from providing detailed comments.
Author Response
Thank you for your valuable suggestions. We have revised the content of our manuscript accordingly. Specifically, regarding the phylogenetic analysis, we have made structural adjustments and deleted some statements about the subfamily discussion. To ensure the main text remains focused on taxonomy, we also streamlined the mitochondrial genome section by relocating Table 2 (Organization of the mitochondrial genome) and Figure 9 (Amino acid frequency and RSCU) to the Supplementary Materials. Additionally, we have included a checklist and an identification key. We believe that these changes better align the paper with its primary focus and enhance the utility of the work for species identification. Please find the specific revisions made in response to the comments below:
Comments 1: A detailed description of a new species, including mitogenomic data, is welcome in principle. However, phylogenetic analysis using a limited and largely random set of species is unlikely to be appropriate in a paper devoted to the description of a single new species; it seems unnecessary and somewhat premature.
Response 1: Thank you for pointing this out. While the current mitogenomic dataset remains limited and the phylogenetic analysis is admittedly insufficient (as noted in the Discussion section), the results nonetheless provide well-supported relationships among congeneric species, providing supplementary evidence for the taxonomic placement of the new species. However, we acknowledge that the extensive emphasis on phylogenetic analysis in the main text was disproportionate to its current scope. After careful consideration, we have deleted some of the statements about the subfamily discussion in the revised manuscript.
Comments 2: In the text, the type species of the genus Nyctegretis is designated as achatinella (page 1), and the distribution is given for lineana (page 2); nowhere is it noted that achatinella is a junior synonym of lineana.
Response 2: Thank you for highlighting this oversight. We have added a species checklist after section 3.1.1, Genus Nyctegretis Zeller, 1848, in which we indicated this synonym relationship. We also revised the Introduction section and moved the distributional data from the Introduction to the Checklist section.
Comments 3: On page 3 (section 3.1.1) the full generic name Nyctegretis Zeller, 1848 is given three times in a row; at least in the second case it is clearly redundant.
Response 3: Thank you. We agree with this. We have removed the two redundant instances of the full generic name Nyctegretis Zeller, 1848 in Section 3.1.1 (Page 3, lines 121-122.)
Comments 4: The specific epithet of the new species is formed incorrectly. Semi- + niger- should be transformed to seminigra or seminigrella (half-black), whereas seminigera in translation from Latin to English means half-breed (half-blood).
Response 4: Thank you for your valuable correction. We have revised the name of the species to “Nyctegretis seminigra” based on your suggestion. Additionally, all scientific names throughout the text have been systematically updated to reflect these revisions.
Comments 5: The article does not provide a decoding of the abbreviations of the depositories of type specimens.
Response 5: Thank you. The expanded names of type repositories are provided in the "Abbreviations" section preceding the References, in accordance with the journal's style guidelines.
Comments 6: The distribution of some Palearctic species of the genus Nyctegretis (lineana, triangulella) is given incompletely, without taking in account published data on different regions of Russia (see Sinev S.Yu., Streltzov A.N., Trofimova T.A. Pyralidae. - In: Sinev S.Yu. (ed.). Catalogue of the Lepidoptera of Russia. Edition 2. St. Petersburg: Zoological Institute RAS, 2019. P. 165-178.)
Response 6: Thank you for pointing out this omission. We have restructured the manuscript to relocate species distributional data from the Introduction to the Checklist section. In the revised version, we added the work of Sinev S.Yu. et al. (2019), and updated the Checklist to include the distribution records of N. ruminella in Russia.
Reviewer 2 Report
Comments and Suggestions for Authors
This is a well written paper with the aim to describe a pyralid moth as new to science, and having supported by a massive genomic elaborational work and its results. Personally I find the genomic technical details too heavy, and would recommend to split the paper into (1) a paper with the description supported by DNA barcode and and mitogenome analysis, and (2) a paper based on nucleotide composition and phylogenic anaylsis. However, this is just my personal bias, and the authors have the right to keep their original intent. Some observations are listed below according to line numbers.
1: The name should be "Nyctegretis seminigrum" if I am right, see Etymology, line 229
13-14: the word "considerable" is used two times too close to each other (not good style)
16: "try to describe" - actually you do not try but you do it, please change it accordingly
36: it is better to write for a genus to be established instead of to be described
48: the listing of countries and geographical locations are confusing, please arrange it according to geography or alpabetically; however it would be more user-friendly to present a check-list of all the species placed in Nyctegretis, what would provide a perfect overview of the diversity; a key for identification would be also handy
123-128: "by original designation and monotypy" when a genus is established via monotypy that means that that action was done by the author, therefore it was original; so just write "by monotypy"
141-153: The Biology and the Remarks entries should be moved after the Etymology section together as Remarks.
228: Before the entry Distribution this brief entry should be add: "Biology. Larval host plant is unknown."
229: The species name is composed using the Latin prefix semi and the Latin noun nigrum. "Niger" is not a Latin word, it is English. I think Nyctegretis is a male noun, so the species group name should be formed accordingly: N. seminigrum. If it turns to be female, in that case N. seminigra. If the authors wants to keep "seminiger", they have to declare, that the word "niger" is a Latinized noun used to signify the black pattern of the species. When Nyctegretis turned to be a female, in that case they can use seminigera. (I have no time to determine the actual gender of the name Nyctegretis.) When the species-group name is changed, the change must be corrected everywhere in the manuscript. Just a side note: according to the documentation I have (Figs 1, 7) the black colour seems to be deep brown.
231: please put here the notes on the biology and nomenclature under the entry remarks.
Author Response
We sincerely appreciate your thoughtful evaluation of our manuscript and your constructive suggestions. We agree that excessive genomic work in the main text was disproportionate to the current scope. After careful consideration, we have adjusted the description of the mitochondrial genome: the two genomic circular maps are now combined into a single figure plate (Figures 7–8); Table 2 (Organization of the mitochondrial genome) and Figure 9 (Amino acid frequency and RSCU) have been moved to the Supplementary Materials. We have also included a checklist and a key to species of Nyctegretis. These revisions ensure greater clarity and alignment with the paper's primary goal of species delimitation and morphological diagnosis. Here are the specific revisions made in response to the comments:
Comments 1: 1 The name should be "Nyctegretis seminigrum" if I am right, see Etymology, line 229.
Response 1: Thank you for your careful review. We have revised the species name to Nyctegretis seminigra in accordance with your suggestion. More details are explained in Comment 9.
Comments 2: 13-14: the word "considerable" is used two times too close to each other (not good style)
Response 2: Thank you. We have changed the second "considerable" to “significant”. On page 1, line 13-14: The considerable diversity and high incidence of homoplasy within Phycitinae present significant challenges for species identification relying exclusively on morphological characters.
Comments 3: 16: "try to describe" - actually you do not try but you do it, please change it accordingly
Response 3: Thank you. We have revised the phrase "try to describe" to "describe". Page 1, line 16: In this study, we describe a new pyralid moth species integrating both morphological and molecular evidence.
Comments 4: 36: it is better to write for a genus to be established instead of to be described
Response 4: Thank you. We have revised the text to use "established" instead of "described" according to your suggestion. Page 1, line 36: The genus Nyctegretis was established by Zeller [1], with Tinea achatinella Hübner designated as its type species.
Comments 5: 48: the listing of countries and geographical locations are confusing, please arrange it according to geography or alpabetically; however it would be more user-friendly to present a check-list of all the species placed in Nyctegretis, what would provide a perfect overview of the diversity; a key for identification would be also handy
Response 5: Thank you for your constructive suggestions. In the revised manuscript, we have added a species checklist of Nyctegretis (in the “3.1.1. Genus Nyctegretis Zeller, 1848” section, page 4, before “3.1.2. Nyctegretis seminigra Yang, Zhou and Ren, sp. nov.”), including their distributional data arranged alphabetically. Accordingly, we removed the previously geographical listings from the Introduction. A key to facilitate species identification was also added after the checklist. These revisions will enhance accessibility and provide a clearer overview of the genus.
Comments 6: 123-128: "by original designation and monotypy" when a genus is established via monotypy that means that that action was done by the author, therefore it was original; so just write "by monotypy"
Response 6: We greatly appreciate your expertise in refining the precision of our terminology. We have revised the phrasing in lines 123–128 by removing "original designation and" and stating simply "by monotypy" as recommended.
Comments 7: 141-153: The Biology and the Remarks entries should be moved after the Etymology section together as Remarks.
Response 7: We appreciate your suggestion but wish to retain the current structure. The "Biology" and "Remarks" sections (lines 141–153) provide genus-level ecological and taxonomic context, so we placed them before the species-level findings. We believe this arrangement keeps genus-related information cohesive, and we hope you understand our decision.
Comments 8: 228: Before the entry Distribution this brief entry should be add: "Biology. Larval host plant is unknown."
Response 8: Thank you. We have added the entry "Biology. Larval host plant is unknown." prior to the "Distribution" section (page 5, line 228) as recommended.
Comments 9: 229: The species name is composed using the Latin prefix semi and the Latin noun nigrum. "Niger" is not a Latin word, it is English. I think Nyctegretis is a male noun, so the species group name should be formed accordingly: N. seminigrum. If it turns to be female, in that case N. seminigra. If the authors wants to keep "seminiger", they have to declare, that the word "niger" is a Latinized noun used to signify the black pattern of the species. When Nyctegretis turned to be a female, in that case they can use seminigera. (I have no time to determine the actual gender of the name Nyctegretis.) When the species-group name is changed, the change must be corrected everywhere in the manuscript. Just a side note: according to the documentation I have (Figs 1, 7) the black colour seems to be deep brown.
Response 9: We gratefully acknowledge your guidance. When establishing the genus Nyctegretis, Zeller derived its etymology from the Greek roots νύξ (nox) and ἐγείρω (excito). The name of Nyctegretis is a Latinized compound derived from Greek roots. In Latin third-declension nouns, names ending in "-is" are generally assigned as feminine gender. Additionally, the type species name (N. achatinella) follows the feminine form. In accordance with your recommendations, we have revised the species name to “Nyctegretis seminigra”, with the etymological root adjusted from “niger” to “nigrum”. All scientific names in the text have been systematically updated to reflect these revisions.
Comments 10: 231: please put here the notes on the biology and nomenclature under the entry remarks.
Response 10: Thank you. However, as mentioned in Response 7, we prefer not to make adjustments here.
Reviewer 3 Report
Comments and Suggestions for Authors
This paper describes a new species from China. It is compared to a closely related species, N. triangulella. Their hypothesis that this is a new species is supported by both morphological and molecular data, including the complete mitochrondrial genomes of both species. They conduct a phylogenetic analysis to show the placement and relationship of these two species within the Phycitinae.
It is the perfect mix of morphological and molecular data to drive robust results. It is well-written and illustrated. The references are accurately applied. Although I chose “The English is fine…” I have noted a few minor issues (mostly grammatical) in the pdf. Also, I would suggest using the term phallus rather than the outdated term aedeagus and suggest using full latin names (rather than anglicized names) for structures, particularly those of the female genitalia. Finally, they included one species of Anerastiini that fell inside the Phycitini. It should be noted in the Discussion. I have suggested verbiage for this result.

Author Response
Comments: This paper describes a new species from China. It is compared to a closely related species, N. triangulella. Their hypothesis that this is a new species is supported by both morphological and molecular data, including the complete mitochrondrial genomes of both species. They conduct a phylogenetic analysis to show the placement and relationship of these two species within the Phycitinae.
It is the perfect mix of morphological and molecular data to drive robust results. It is well-written and illustrated. The references are accurately applied. Although I chose “The English is fine…” I have noted a few minor issues (mostly grammatical) in the pdf. Also, I would suggest using the term phallus rather than the outdated term aedeagus and suggest using full latin names (rather than anglicized names) for structures, particularly those of the female genitalia. Finally, they included one species of Anerastiini that fell inside the Phycitini. It should be noted in the Discussion. I have suggested verbiage for this result.
Response: We deeply appreciate your recognition of our work and constructive suggestions. We have carefully addressed nearly all of your recommendations. Specifically, we have updated the terminology for female genitalia structures to their full Latin names and adjusted the corresponding figure to reflect these changes. We have also included a checklist and a key for identification. Additionally, all grammatical issues noted in the PDF have been corrected. Below is a summary of the main revisions:
- We have revised "bronzy" to "bronze". (page 5, line 191)
- In the description of female genitalia, we have revised “papilla analis” to “papillae anales” (page 5, line 217), “apophysis anterior” to “apophysis anterioris”, “apophysis posterior” to “apophysis posterioris”, “ostium” to “ostium bursae” (page 5, line 220).
- The female terminology in "Figures 1–6." has been revised, and the images have been replaced. (page 6, ling 232)
- We have revised “It is noteworthy, however, that these studies did not comprehensively sample most taxa of Phycitinae, focusing mainly on species from Phycitini and Anerastiini (only one species).” to “It is noteworthy, however, that these studies did not comprehensively sample most taxa of Phycitinae, focusing mainly on species from Phycitini. Only one species of Anerastiini (Peoria approximella) was included in the analysis and it is nested within Phycitini.”
However, after careful consideration, we have decided to retain the terms of "wingspan" and "aedeagus" in this context. While wing length is indeed a valuable measurement, we chose to use wingspan in our description to maintain consistency with the majority of literature on this genus. We acknowledge that “phallus” aligns more closely with unified anatomical terminology, but we have kept the term “aedeagus” in this study to adhere to the descriptive conventions commonly used for the taxonomic group. We hope that you can understand our decision.